# The Hulks and the Deadpools of the Cytokinin Universe: A Dual Strategy for Cytokinin Production, Translocation, and Signal Transduction

**DOI:** 10.3390/biom11020209

**Published:** 2021-02-03

**Authors:** Tomáš Hluska, Lucia Hlusková, R. J. Neil Emery

**Affiliations:** 1Laboratory of Hormonal Regulations in Plants, Institute of Experimental Botany, The Czech Academy of Sciences, Rozvojová 263, CZ-16502 Prague 6, Czech Republic; hluskova@ueb.cas.cz; 2Biology Department, Trent University, Peterborough, ON K9L 0G2, Canada; nemery@trentu.ca

**Keywords:** *cis*-zeatin, cytokinins, cytokinin biosynthesis, isopentenyl transferase, cytokinin oxidase/dehydrogenase, cytokinin signalling, aromatic cytokinins, cytokinin transport, Hulk/Deadpool

## Abstract

Cytokinins are plant hormones, derivatives of adenine with a side chain at the *N^6^*-position. They are involved in many physiological processes. While the metabolism of *trans*-zeatin and isopentenyladenine, which are considered to be highly active cytokinins, has been extensively studied, there are others with less obvious functions, such as *cis*-zeatin, dihydrozeatin, and aromatic cytokinins, which have been comparatively neglected. To help explain this duality, we present a novel hypothesis metaphorically comparing various cytokinin forms, enzymes of CK metabolism, and their signalling and transporter functions to the comics superheroes Hulk and Deadpool. Hulk is a powerful but short-lived creation, whilst Deadpool presents a more subtle and enduring force. With this dual framework in mind, this review compares different cytokinin metabolites, and their biosynthesis, translocation, and sensing to illustrate the different mechanisms behind the two CK strategies. This is put together and applied to a plant developmental scale and, beyond plants, to interactions with organisms of other kingdoms, to highlight where future study can benefit the understanding of plant fitness and productivity.

## 1. Introduction

Cytokinins (CKs) are plant hormones involved in many physiological processes. They are considered as one of the main groups of phytohormones as they, together with auxins, control cell division and, hence, influence the overall plant’s architecture. 

The first CK was discovered in the mid 1950s [1]. Its cytokinetic activity was detected from old and autoclaved DNA and the purified compound was named kinetin. This spurred a search for compounds, natural and synthetic, with similar activity. Interestingly, the first compound with CK-like activity isolated from a natural source (that is, apart from degraded DNA) was *N*,*N′*-diphenylurea [2]. It was isolated from coconut milk, as that was previously recognised to induce cytokinesis. The coconut milk was, in fact, used in growth media before the discovery of CKs. However, the identified *N*,*N′*-diphenylurea was probably present because of contamination as the coconuts were processed in a facility where urea herbicides had been previously synthesised [3]. To date, the presence of *N*,*N′*-diphenylurea in the coconut milk has never been confirmed. On the other hand, the presence of the kinetin in coconut milk was validated [4]. However, kinetin was implied to originate by oxidative damage to DNA bases [5] and this is consistent with its discovery in old and autoclaved DNA; thus, it is likely not synthesised *de novo* as per other CKs.

Thus, the first unambiguously identified natural CK was zeatin [6], presumably the *trans* isomer, as the purification was followed by a bioassay for cell division (see below).

There is a hypermodified nucleotide in many tRNAs forming A=U pairing at the first position of the codon. While the tRNAs recognising UNN codons contain mostly CKs or wybutosine, a hypermodified guanine, those recognising ANN codons contain almost exclusively *N^6^*-threonylcarbamoyladenosine. No such conservation is present in tRNAs binding to CNN or GNN codons [7].

The modifications in the anticodon loop of tRNA generally serve to improve codon-anticodon reading and increased translation efficiency. The hypermodifications of purine at position 37 (*N^6^*-isopentenyl, *N^6^*-threonylcarbamoyl and/or 2-methylthiol groups) facilitate the complementary codon binding [8]. In this way, it contributes to the modified wobble hypothesis [9]. 

CKs are present in tRNAs of organisms from all life kingdoms with the possible exception of Archea [10]. The CK is present at position 37 (A37), next to the anticodon. The modification occurs at the middle one of three adenosines. This is not the sole requirement though, as there are other unmodified adenosine triplets [11]. However, CKs are not present in all tRNAs of this type or even consistently in analogous tRNAs from different organisms [7]. 

The functions of CKs can be summarised as triggering cellular changes that are essential for numerous possible fates throughout the plant’s life. This includes both developmental processes and adaptive responses to various abiotic and biotic inputs. The response is further influenced by crosstalk with other signals that vary with the cell’s history and thus context [12].

## 2. Hulks and Deadpools of the Cytokinin Universe

In this paper, we are proposing a novel hypothesis describing a two-tier system of CK types, their metabolic enzymes, and signalling components to regulate plant growth and to distinguish between times of prosperity and times of compromise.

In the pop culture, one can encounter distinct Universes in which there are certain physics and figures who co-exist and may meet. In the Marvel Comics Universe, there are, among many others, Hulk who is incredibly strong and Deadpool who is immortally resilient (Figure 1). Both characters have superhuman powers. However, they have strikingly different abilities and roles. Hulk is a hyper-muscular humanoid possessing a vast degree of physical strength but only appears for relatively short-lived bursts of angry power while Deadpool is not extraordinarily physically strong but has immortal persistence derived from accelerated healing factors. 

Throughout the Universe of the plant hormones known as the CKs, there seems to be a reoccurring theme of the strong but specialised “Hulk” versus the less active but more resilient “Deadpool” (Figure 1). Below we will illustrate the key differences with a few examples that will be corroborated in more detail in subsequent sections. 

The activity of particular CKs may be described at two levels. First, CKs influence various physiological processes and this can be quantified in various bioassays. Second, CKs bind and activate their receptors, which is usually measured in heterologous systems. These two measures sometimes do not overlap. For example, *O*-glucosides or ribosides exert CK activity in various bioassays (e.g., [7]), but *O*-glucosides do not bind to receptors at all and ribosides very weakly [13,14]. They are active because they can be metabolised to the active forms. On the other hand, *N^6^*-(Δ^2^-isopentenyl)adenine (iP; all CK abbreviations are in accordance to Kamínek et al. [15]; see Appendix A) binds strongly to receptors, but it is not effective in preventing senescence in monocots (e.g., [16,17]).

Furthermore, the judgement about whether a particular CK exerts its activity through binding to CK receptor(s) is often reduced to comparisons of dissociation constants. However, even compounds with (relatively) higher dissociation constants may bind to the receptors even at (relatively) low concentrations. Of course, under such conditions, the receptors will not be saturated, but there will be certain output anyway, albeit suboptimal. The second factor, which is currently overlooked, is the duration of the signal. In this matter, a long-lasting weak signal may result in significant output.

On the one hand, there are the highly active Hulk CKs, *trans*-zeatin (*t*Z) and iP, which are capable of inducing a burst of growth, and levels of which are tightly controlled by cytokinin oxidase/dehydrogenase (CKX) or other deactivating enzymes. On the other hand, there are Deadpools, such as *cis*-zeatin (*c*Z) and aromatic CKs, which are, in general, weak substrates, and thus less likely to be destroyed by CKX [18,19]. However, this resistance/lower vulnerability to deactivation makes the feeble signal of the Deadpool CKs persistent (i.e., as demonstrated for *Rhodococcus*; [20]). It may be that the function of *c*Z in stress signalling is to simply “knock-out” the interaction of the strong Hulk CKs with enzymes of CK metabolism or with receptors. This would have a dual effect. By competing for binding to receptors, it may diminish CK signalling during stress until more favourable conditions occur; while there would also be competition for binding to inactivation enzymes, such as glycosylase or CKX, and in effect it would prevent degradation or conjugation of the Hulk CKs. This may be favourable for said *Rhodococcus*.

The apparent strength of respective enzymes, receptors, or transporters can be similarly judged by comparisons of their activities and the corresponding *in planta* spatio-temporal expression patterns. Accordingly, the Hulks are expressed and/or localised in a more focused matter and exert high activity, while the Deadpools have more widespread expression with rather low activity, which consequently yields more uniform output.

For example, the expression of *AtIPT*3 is strongly upregulated upon nitrate re-supply as a signal for fast growth while the resources are available. On the other hand, the expression of *IPT*5 orthologues is more uniform and their activity is rather low [21,22]. Thus, although they produce *t*Z and/or iP [23], they provide a more constant supply of CKs. Indeed, *AtIPT*5 expression correlates with nitrogen concentration in media when supplemented continuously [24]. 

Next, the Lonely guy (LOG) proteins are capable of quickly and specifically activat-ing CKs (almost) directly from their biosynthetic forms [25] and thereby act as Hulks; yet, there is the possibility of two-step activation (de-phosphorylation followed by de-ribosylation), which may originate simply from a plant’s inability to regulate any non-specific activity. However, as discussed below, while this Deadpool two-step activation may be less responsive, it nonetheless retains physiological significance. 

Likewise, the CK receptors on the plasma membrane perceive long-range Hulk signals, while the ER localised receptors likely perceive mainly the autocrine Deadpool signal. Interestingly, recently described CK receptors from apple trees (*Malus domestica*), *Md*CHK3a and *Md*CHK3b, show constitutive activity, which may be enhanced by added CKs [14]. Their expression was detected in all tissues tested, but it was about 100-times lower than of *Md*CHK2, the most active and expressed member of the CK receptor family in apple trees. This would be an exemplary case of Deadpool signal transduction. There also appears to be low- and high-affinity transport of CKs in plants, as the CK uptake rates show multiphasic characteristics [26]. 

Thus, to expand the previously published hypothesis that *c*Z is important under growth-limiting conditions [27,28], we are presently proposing an extension of that hypothesis, whereby there are two sets of metabolites, enzymes, transporters, and percep-tion relays for CKs (Table 1). While one, including but not limited to *t*Z, iP, *At*IPT3, and *At*LOG7, is highly active and has specialised functions in promoting and regulating rapid growth (“Hulk”), there is another including but not limited to *c*Z, aromatic CKs, and *At*LOG8 that is involved in maintenance growth (“Deadpool”). 

This hypothesis was formulated based mostly on data from *Arabidopsis*, because most of the foundational data come from that species. While enzyme, receptor, and transporter orthologues will probably work similarly (i.e., could be classified as Hulk or Deadpool in analogy to their *Arabidopsis* counterparts), this does not have to be the case for metabolites. For example, certain plants contain predominantly *cis*-zeatin. In such plants, it may function more as a Hulk CK as discussed below. Thus, it is possible that in other organisms, certain metabolites, but possibly also enzymes, transporters, or receptors, would be categorised differently than in *Arabidopsis* (Table 1).

Furthermore, the categorisation of some features may shift based on the magnitude of “scale”. For example, in the CK activation landscape, the LOGs would be generally perceived as Hulks; however, *At*LOG8 has the weakest activity among them, yet its expression level is also the highest [25]. Therefore, it substantially activates small amounts of CKs throughout the plant similarly to the two-step activation pathway.

This Hulk/Deadpool two-tier system could reflect the layers of growth strategies, powerful, short-lived, and focused on the one hand and persistent and pervasive on the other hand. It furthermore serves the source/sink relationships among plant organs, which sometimes necessitate rapid growth for fecundity (i.e., filling seeds) but cannot come at a cost to critical but slower growing spots (i.e., vegetative tissue maintenance). Overall, it reflects the original defining characteristics of CKs, which were primarily discovered for their cytokinetic properties.

## 3. Cytokinin Types

Cytokinins are a family of compounds derived from adenine with a side chain at the *N^6^*-group. This side chain defines two groups of CKs (Figure 2). Modifications of the side chain further define particular CK types and their metabolism and biological activity.

### 3.1. Cis or Trans—That’s What Matters

The first CK isolated from a natural source was zeatin obtained from immature seeds of sweet corn (*Zea mays*, hence the name; [6]). As the hydroxymethyl group is positioned at a double bond, zeatin forms two geometric isomers. Quickly, it was established that the highly active compound was *trans*-zeatin. Soon, *cis*-zeatin was identified in tRNA [29]. However, its activity in the classical bioassays was miniscule [30,31,32]. Additionally, the activity of several enzymes of CK metabolism (especially of the catabolic CKX) towards *c*Z was often negligible and thus *cis*-zeatin was often neglected.

Over the years, however, the number of reports of plants or plant organs containing predominantly the *c*Z-type CKs has kept growing. This includes potatoes [33,34,35,36], unfer-tilised hop cones [37], rice [38,39], chickpea and white lupine [40,41], maize [19,42,43], *Tagetes minuta* [44,45], pea [46], *Lolium rigidum* [47], lucerne and oats [48], oilseed rape [49], and Norway spruce somatic embryos [50]. The large-scale analyses in the plant kingdom [27], bryophytes [51], and in cyanobacteria and algae [52] revealed that the *c*Z dominance is not a unique trait. Further, in other kingdoms of life, such as fungi, *c*Z is often the major CK type [53]. 

Beyond their frequent predominance there are many examples that *c*Z types can alter growth and development. For example, plants lacking *c*Z display altered phenotypes, which suggests the significance of *c*Z *in planta*. This may be achieved either through decreased *c*Z biosynthesis or its increased degradation. Thus, *Arabidopsis* with decreased production of *c*Z were often chlorotic [54] and the leaves of plants with increased *c*Z degradation displayed visible stress symptoms and accumulation of anthocyanin [55]. Moreover, both types of plants showed aberrant root development with a shorter primary root [55]. This is in strong contrast to the increased primary root length of other lines with decreased levels of iP and *t*Z [54,56]. It must be noted that the lines with decreased *c*Z biosynthesis are mainly affected in the levels of CKs in tRNA and the plants with increased *c*Z degradation also have decreased levels of other types of CKs. However, *c*Z was shown to complement the root developmental abnormalities, similarly to *t*Z, albeit at higher concentrations [55]. Moreover, *c*Z shows activity in systems that are relevant to it (i.e., where they are at high levels). For example, the development of pea embryos is promoted by *c*Z [46]. *c*Z was also shown to inhibit the growth of roots of *c*Z-dominant rice, a response that is considered typical for CKs [38].

The low activity in bioassays and low content in some plants might suggest that *c*Z is indeed only an artefact from tRNA hydrolysis, without physiological significance, and that the interaction with CK enzymes is merely because of their low ability to discriminate between the isomers. When *c*Z was found to be active in some bioassays, it was usually explained by isomerisation to the *trans*-isomer. However, the discovery of the first enzyme highly specific towards *c*Z [57] has unambiguously shown that its levels must be regulated by the plant and it is, thus, physiologically relevant. Additionally, the direct interaction of *c*Z with CK receptors [58,59] showed that it is solely responsible for its own activity. Even though the majority of CKX enzymes degrade *c*Z inefficiently, there are few isoenzymes that react with *c*Z, and some of them even preferentially over the other CKs [19,27,55]. Thus, if *trans*-zeatin were the “big and strong brother” of the Zeatins family, the *cis*-zeatin would be the, maybe weaker, yet more resilient brother.

A possible objection to the idea that *c*Z is synthesised mainly or exclusively via tRNA is that it would be a waste of energy and resources for the plant (in terms of the synthesis of tRNA for the production of significant amounts of hormone). However, as the plant tRNA contains mainly *cis*-zeatin (see below and [54]), which is the main CK involved in stress responses (as reviewed by Schäfer et al. [28]), it is logical that a specific compound (i.e., not unmodified nucleotides) released from RNA can serve as a stress signal because the RNA hydrolysis is stress sensitive [60]. In this manner, *c*Z works downstream of some primary response, which causes the RNA hydrolysis. Possibly, the sole function of *c*Z in stress signalling may be simply to “knock-out” the strong Hulk signal by competing on CK receptors, until more favourable conditions occur.

Moreover, regular RNA turnover keeps the level of *c*Z, which may serve as the maintenance Deadpool signal. Furthermore, *c*Z has an advantage to serve as both a stress and maintenance signal, because the plant already possesses a response system that slightly cross-reacts to it, unlike any other modified RNA bases. 

Since the relevance of tRNA-derived CKs is debatable, it was proposed that CKs were first present in tRNA for translation efficiencies, and only later acquired functions as hormones [61]. Plants needed to avoid interference of their developmental processes with tRNA turnover and this may have been achieved in mainly three ways [62]:Reduction of tRNA species bearing CK-like modifications and replacement with others, such as wybutosine.Preference for *cis*-zeatin over iP in tRNA.Modulation of CK perception to prefer *trans*-zeatin.

Thus, while bacteria contain CKs in most of their tRNAs recognising UNN codons, either as iP or *c*Z, depending on the presence of the *Mia*E gene (cf. *Escherichia coli* and *Salmonella typhimurium*), plants’ tRNAs contain relatively few CKs and the majority is present as *c*Z [54].

The importance of the tRNA-derived CKs has been demonstrated only recently. The *Bradyrhizobium* sp. strain ORS285 *mia*A mutant showed delayed development of nodules on *Aeschynomene* plants [63]. Similarly, the *cptrna ipt* mutant showed decreased virulence. Formation of sclerotia, the dormant stage of the fungus, was retarded and the majority of sclerotia were white and soft [64].

### 3.2. Dihydrozeatin

The reduced form of zeatin—dihydrozeatin (DHZ)—is considered less active than *trans*-zeatin [7]. Yet, it was first isolated from yellow lupine immature seeds following activity measurement using the tobacco callus bioassay [65,66] and showed a substantial activity in other works [30,31]. In fact, it was the most active isoprenoid CK in a *Phaseolus vulgaris* bioassay [30]. In the same assay, isopentyladenine, a saturated analogue of iP, was 100-times more active than its parent compound. Further, it is capable of activating at least some CK receptors, especially the HK3 orthologues [58,67,68,69,70].

Despite its potential importance, DHZ is omitted from many analyses and hypotheses nowadays (e.g., [23,54]). This is probably because of its generally low measured quantities and because of a complete lack of knowledge about its metabolism. For example, there are only three publications dealing with the zeatin reductase, the putative DHZ biosynthetic enzyme, published over the course of 35 years [71,72,73]. We do not even know how DHZ is removed from the plant, as it is resistant to degradation by CKX [18]. Conversion of DHZ to adenine and/or adenosine was reported only rarely [74,75]. Otherwise, there is no evidence of other conversions besides glycosylations. 

Dormant seeds often contain CKs with lower activity, such as *cis*-zeatin [27,40,46,48] or dihydrozeatin [19,40,48,76]. There seems to be a pattern of *t*Z-dominant plants having *c*Z in their seeds and *c*Z-dominant plants having DHZ in their seeds, although more widespread analysis would be necessary. 

All of this together positions DHZ at the borderline of Hulks and Deadpools. It is highly active in bioassays, hence acting as Hulk, but only in a few known processes. Moreover, it is probably produced only at low rates. Furthermore, it is resistant to deactivation and thus acting as Deadpool, at least in *Arabidopsis*. Therefore, DHZ is an adjunct in the Zeatins family that only few researchers are talking about.

Lately, the authors have encountered a few times the notion that DHZ is not isoprenoid CK because of the saturated side chain. Although the information on zeatin reductase is scarce, as long as we do not have evidence for another origin of DHZ, we must assume that it originates from *t*Z reduction and thus belongs to the isoprenoid CKs.

### 3.3. Aromatic Cytokinins

Aromatic cytokinins are adenine derivatives with an aromatic side chain (Figure 2). They include *N^6^*-benzyladenine (BAP) and its hydroxyderivatives called topolins and their methoxyderivatives [77,78], *N^6^*-furfuryladenine (kinetin; [4,79]), and the recently (re-)discovered 6-(3-methylpyrrol-1-yl)purine (MPP) and its derivatives [80,81,82]. 

Unlike the metabolism of isoprenoid CKs that has been extensively studied, the metabolism of aromatic CKs, especially their biosynthesis, is largely unknown. This is partly because of their scarce occurrence and hence lack of a suitable model for their study. So far, they are known to be very weak substrates for the CKX [18,83]. This is why they are very effective as morphogens in various in vitro culture techniques. While *meta*-topolin mimics *t*Z both in *O*-glucosylation reaction and receptor recognition, *ortho*-topolin similarly mimics *c*Z [84]. 

Since poplar trees are known to synthesise aromatic CKs, there was an effort to see if poplar isopentenyl transferases (IPTs) were special in that regard. It was concluded that the IPT enzymes are not responsible for biosynthesis of BAP-derived CKs because *Arabidopsis* transformed with poplar *IPTs* did not contain any aromatic CKs [22]. Nonetheless, when produced in *Arabidopsis*, the poplar enzymes may have simply lacked the appropriate substrate for the synthesis of aromatic CKs. Paradoxically, unambiguous information about biosynthesis exists only for the newest aromatic CK—MPP. It is synthesised directly from *trans*-zeatin, although the biosynthetic enzyme was not purified nor identified due to its instability [81]. In this regard, its origin from *t*Z makes it interestingly in fact an isoprenoid CK. 

It has been shown that MPP is capable of binding both to CK receptors and to CKX, although it neither activates them nor is it a substrate of CKX [81]. Thus, it has a dual role. It decreases the response of CK receptors; but, by inhibiting CKX, it extends the CK response. As MPPR and MPP9G are present in amounts comparable to other CKs, their effect on CKX may be even more substantial as ribosides and *N9*-glucosides are often preferred substrates of CKX. On the other hand, they will not affect the receptors, which normally do not bind *N9*-glucosides and only weakly bind ribosides. However, MPPR and MPP9G probably will not be able to bind to CKX in the same unique way as was reported for MPP. So, the question remains: would they be able to inhibit CKX?

In addition to the *c*Z- and DHZ-types, plant seeds are often reported to contain aromatic CKs. This holds for examples like kinetin found in the liquid endosperm of coconut (for a review, see [85]), MPP in the seeds of maize [81], and BAP and topolins in *Tagetes minuta* [44,45], pea [86], the seeds of maize, oat and lucerne [48], and in many other plant species [86]. This suggests that aromatic CKs may perform a common role in seed germination.

The natural occurrence of aromatic CKs has been shown in only a few select plants, algae, and microorganism species [22,45,48,77,78,87,88]. For example, derivates of *ortho*-topolin were identified by at least four independent groups in various plants [22,89,90,91]. However, in many cases, their detection in living samples might be the result of lab-ware contamination due to their massive usage in in vitro culturing. It was hypothesised that the aromatic CKs in plants may in fact originate from symbiotic bacteria. In accord with this hypothesis, BAP was reported to be secreted by nematodes [92] that are known for their intimate interactions with bacteria [93]. The bacterial origin would explain the inconsistency in the detection of aromatic CKs in various plants. 

Another option is that aromatic CKs may originate from nucleic acids. For starter, *ortho*-topolin was reported in the form of 2-methylthio-derivative [90]. The 2-methylthio-CKs are currently thought to originate solely from tRNA hydrolysis. Further, one laboratory has identified *ortho*-topolin directly in the RNA of algae *Chlorella sorokiniana* [94] and poplar [22]. Interestingly, the content of aromatic CKs was shown by some of the same researchers to change during daytime [22,78]. 

## 4. Cytokinin Biosynthesis

The biosynthesis of CKs starts with the addition of a side chain to an adenine either in nucleotide form or bound in tRNA. Several additional steps are generally included in the CK biosynthesis as hydroxylation of the side chain or reduction of the double bond in *trans*-zeatin’s side chain (Figure 3).

The most recent research has been focused on *Arabidopsis* and on iP and *t*Z as the main active CKs [95] and, thus, there has been a significant bias to their reporting. As has been mentioned above, the synthesis of DHZ from *t*Z has received very little attention so far and the *de novo* biosynthesis of aromatic cytokinins remains an enigma. Another example is the assumption of strict preference of the precursor’s biosynthetic pathway used for synthesis of iP and *t*Z or *c*Z, when in fact the majority of living organisms have only one biosynthetic pathway operational with the exception of green plants and only a handful of microorganisms [96,97]. Yet, although fungi, animals, and some bacteria do not possess the methylerithrytol pathway purportedly responsible for iP and *t*Z precursor synthesis and/or they lack enzymes for *de novo* CK synthesis, most of them produce the full spectrum of CKs.

### 4.1. Isopentenyltransferase

The first step of CK biosynthesis is catalysed by isopentenyltransferases. The correct name is dimethylallyltransferase, as dimethylallyl pyrophosphate (DMAPP) is the substrate, rather than Δ^3^-isopentenyl diphosphate (IPP). However, throughout this work, the commonly used name isopentenyltransferase (IPT) is used.

The addition of the isoprenyl side chain is widely referred to as the rate-limiting step of CK biosynthesis [98]. Considering that the free bases are usually present in minute quantities while the ribosides and ribotides are often present in much higher quantities, it seems probable that another step in the CK biosynthesis/activation process is the bottleneck.

There are currently three classes of isopentenyltransferases recognised to be involved in CK biosynthesis: adenylate dimethylallyltransferase (AMP-dependent; EC 2.5.1.27), tRNA dimethylallyltransferase (EC 2.5.1.75), and adenylate dimethylallyltransferase (ADP/ATP-dependent; EC 2.5.1.112). At this moment, only the tRNA IPTs seem to strictly adhere to the single activity of dimethylallyl pyrophosphate:adenine^37^ in tRNA dimethylallyltransferase. The adenylate IPTs, on the other hand, are enzymes utilising multiple substances both as acceptor and as donor substrates. 

In plants, there are two pathways providing isoprenoid units with distinct localisation and usage of the products. The first one is the mevalonate pathway (MVA), which operates in the cytosol of plants and is present also in animals, fungi, Archea, some bacteria, *Trypanosoma*, and *Leishmania*. The end-product is IPP, which can be freely isomerised to DMAPP. No hydroxylated intermediate was found to date in this pathway. The second pathway is called the methylerythritol pathway (MEP) or non-mevalonate pathway and operates in the plastids of plants, in green algae, and in the majority of bacteria. A series of steps leads to (*E*)-4-hydroxy-3-methylbut-2-enyl diphosphate (HMBDP), which is ultimately reduced to yield either IPP or DMAPP. There is an exchange of IPP and DMAPP among the compartments of the plant cell to some extent [99]. DMAPP and HMBDP serve as side-chain donors in iP and *t*Z biosynthesis, respectively [100].

In reports concerned more with the IPT enzymology than the CK function, a wider substrate specificity was reported, including mono-, di-, and tri- nucleotides of purine and pyrimidine bases [101,102]. Interestingly, the *Hl*IPT from hop is capable of binding diadenosine polyphosphates and using them as substrates. It is even capable of forming diisoprenylated product [103]. *Ma*IPT from mulberry prefers the DMAPP as the donor substrate, but it is capable of also using HMBDP, geranyl pyrophosphate (C_10_), and IPP. Farnesyl pyrophosphate (C_15_) is not an acceptable substrate for *Ma*IPT [101]. The *Hl*IPT possesses a large cavity, which could also able to accommodate geranyl pyrophosphate [102]. Of course, one can neglect this as an artefact without physiological significance. However, nonetheless, it would be interesting to see whether, for example, enzymes from *Arabidopsis* would be able to utilise similar substances as substrates and/or whether mulberry or hop contain CK-like compounds derived from these substances. Afterall, *N^6^*-farnesyladenine and *N^6^*-geranyladenine displayed activity in a tobacco callus bioassay [104].

Although it was assumed in the past that CKs are synthesised in the root and transported to the aerial organs, expression analysis showed that *IPT* expression is widespread throughout the whole plant [24,43,105,106,107,108,109,110]. It is also responsive to other hormones, mostly auxin [105,107,108,111,112], and to some nutrients, mostly to nitrate [24,105]. Recently, it was shown how nitrate promotes plant growth through *IPT* expression and CK production [113,114].

Besides differential expression, the proteins are localised to various subcellular compartments. With the exception of cytosol-localised *At*IPT4 and mitochondria-localised *At*IPT7, the remaining *Arabidopsis* adenylate IPTs were shown to be localised to plastids [23]. However, this result was based on C-terminal fusion with green fluorescent protein (GFP). When the GFP was fused to the N-terminus, *At*IPT3 was localised to the nucleus due to farnesylation [115]. A non-farnesylable C333S mutant was placed in plastids, unless it lacked the chloroplast transit peptide. In this case, it was localised in the cytoplasm [115]. Similarly, the closest homologue in poplar, *Pc*IPT3, is the only poplar IPT containing CLVA peptide [22]. It was hypothesised that the dual localisation may be a regulation dependent on the relative availability of isoprenoid units coming from MVA and MEP pathways [115].

The kinetics of IPT enzymes have never been studied in as much detail as, for example, those of the CKXs and thus the importance of the different isoforms is difficult to judge. However, the Hulk *At*IPT3 clearly stands out with its strong upregulation upon renewal of nitrogen [24,105] serving as a signal for fast recovery of growth. Intriguingly, the double localisation of *At*IPT3 may be the mechanism that ensures sufficient amounts of substrate for production of the Hulk signal. On the other hand, expression of *AtIPT*5 positively correlates with nitrogen availability in the long run [24]. In general, expression of *IPT*5 orthologues, *AtIPT*5, *SlIPT*3, and *PpIPT*5a, is widespread [21,22,105], although it is low in some cases and the enzyme activity is rather low. As a matter of fact, expression of *AtIPT*5 and 7 is upregulated by auxin [105], which generally counteracts CKs. Thus, *IPT*5 orthologues seem to be the omnipresent Deadpool signal indicating sufficient nutrients for stable growth and balancing of auxin action. Similarly, tRNA IPTs are, at least in *Arabidopsis*, responsible for the production of the Deadpool *c*Z [54]. Hence, they could be classified as Deadpool. Indeed, they are usually widely expressed (e.g., [43,105]).

### 4.2. Hydroxylation of Isopentenyladenine

In the iP-dependent pathway, zeatins are formed by hydroxylation of the side chain.

This was first observed in the culture of the fungus *Rhizopogon* by Miura and co-workers [116,117]. Later, such activity was observed in microsomal fraction from cauli-flower [118]. NADPH-dependent microsomal proteins converted iP riboside (iPR) to *t*Z riboside (*t*ZR) and iP to *t*Z. It was deduced, that cytochrome P450 monooxygenase (CYP) is responsible for the reaction, as it was inhibited by metyrapone, ethylene, and CO [118]. Later, two such CYPs were identified in *Arabidopsis* [119]. The enzymes utilise nucleotides of iP, mainly as mono- and diphosphates. Free base and riboside were very weak substrates. The only products detected in vitro were *t*Z-type CKs. Interestingly though, *Saccharomyces cerevisiae* expressing *AtIPT*4 with the cytochromes also contained a significant amount of DHZ-type CKs. 

Both genes are expressed mainly in the roots and their expression is upregulated by iP and *t*Z and downregulated by auxin and abscisic acid [119]. It was confirmed that the *At*CYP735As are major contributors to *trans*-zeatin synthesis, since a double knock-out mutant had <5% of *t*Z-type CKs, while iP-type CKs were increased to ~200% [120]. The mutant’s phenotype resembled the phenotype of other CK-deficient or CK-insensitive mutants *ipt*3 5 7 and *ahk*2 3. Although the plant’s growth was affected mainly in the shoot and barely in the root [120], it showed defects in lateral root primordia positioning [121]. The phenotype could be rescued by supplementation of *t*Z, but not iP or DHZ, thus reflecting the lack of *t*Z as the causative agent of the phenotype [120].

### 4.3. Cis-Zeatin—Unde es?

Unlike the biosynthesis of *t*Z and iP, the biosynthesis of *cis*-zeatin remains a mystery. Nowadays, it is accepted that *c*Z originates from tRNA [54]. However, that does not answer the question—where does the *cis*-hydroxylated side chain originate from? There are, in principle, four possibilities for *cis*-zeatin biosynthesis:(1)Isoprenylation of adenine, either free or tRNA-bound, with (*Z*)-4-hydroxy-3-methylbut-2-enyl diphosphate.(2)*cis*-hydroxylation of iP, either free or tRNA-bound.(3)Isomerisation of *t*Z.(4)Dehydrogenation of dihydrozeatin.

The first two options are analogies to *t*Z biosynthesis. A *cis*-hydroxylase is known to act upon iP in tRNA [122]. The gene was named *Mia*E in *Salmonella typhimurium*. In general, the gene is present only in few species, such as *Nostoc* [123]. However, no *Mia*E homo-logue has been identified yet in plants. Thus, there is no evidence for either of the two options to function *in planta* at the moment. 

Zeatin *cis-trans* isomerase was described [124], but neither the gene nor protein were identified. Thus, it was mentioned only when the discrepancy in data was floppily explained [59,82,125,126,127], but its existence was recently disproved [128]. Indeed, a majority of the *in planta* evidence is against the existence of zeatin *cis-trans* isomerase [23,27,54,59,129,130], with the exception of a few cases involving applications of high CK concentrations whereupon the plant does not manage to control the metabolism, opening the possibility for non-enzymatic reactions [22,34,38,81,131].

The last option is merely a hypothetical possibility; no such enzyme has ever been reported. However, dihydrozeatin is resistant to CKX cleavage [18,19,132], but it disappears during seed imbibition and following seedling growth. Thus, it was hypothesised that DHZ could serve as a source of active CKs before *de novo* biosynthesis starts [133]. Indeed, there was an attempt to measure the putative dihydrozeatin oxidase. Although no *t*Z or *c*Z was detected, a substantial amount of adenine was formed as a consequence of substrate degradation [134]. Accordingly, Podlešáková et al. [75] reported conversion of DHZ9G to adenine in maize.

Thus, *cis*-zeatin biosynthesis remains a big unknown in the CK world.

## 5. Activation of Cytokinins

The CKs are synthesised as nucleotides, but free bases are the active forms. Thus, hydrolysis of phosphates and—optionally—of ribose must precede any biological activity.

Enzymes with wide substrate specificities, such as 5′-nucleotidase, or alkaline and acid phosphatases, were thought to dephosphorylate the nucleotides. Additionally, the enzymes of the nucleotide salvage pathway may be of importance to CK metabolism (as reviewed by Chen [135]). 

A CK-specific phosphoribohydrolase has been identified in rice [136]. A mutant of the corresponding gene in rice was identified based on a screen for plants with defects in the shoot meristem. Flowers of this particular mutant often contained only one stamen and no pistil. Hence, it was called *lonely guy* (*log*). Because of a clustering of LOG homologues in *Agrobacterium rhizogenes* and *Rhodococcus fascians* to *IPT* genes, its involvement in CK metabolism was elaborated. Indeed, it was shown to activate the CK nucleotide monophosphates in one step. None of the following underwent conversion: CK di- or triphosphates, AMP, CK nucleosides, or free bases [136]. Similar results were obtained with LOG from the pathogenic fungus, *Claviceps purpurea*; although, low activity with AMP was reported [137]. Notably, the activity with AMP was higher than with di- and triphosphates of CKs. Bacterial LOGs [138,139,140,141], as well as Archeal LOG [142], were shown to hydrolyse AMP. Their activity towards CK nucleotides was not estimated in vitro, with the exception of the *Mycobacterial* enzyme. 

Notably, no *IPT* genes were identified in the genomes of Archaea [143] and so they are not expected to contain any CKs. Thus, the function of the LOG-like proteins in Archaea may be sensing and/or deactivation rather than activation of signalling compounds, in this case possibly cyclic di- and linear oligo-nucleotides [144].

The rice and *Arabidopsis* genomes contain 11 and 9 *LOG* genes, respectively [25,136]. The *Arabidopsis* proteins in general do not discriminate much between respective nucleo-tide monophosphates of isoprenoid CKs. Activity with BAPRMP ranged approximately from 5% to 75% of the activity relative to that with iPRMP [25]. Upon *LOG* overexpression, plants showed increased levels of iP, iP7G, and iP9G; decreased levels of iPRMP; and levels of iPR remained the same. 

*AtLOG*7 possesses a central role in the CK activation, as knockout of only this one gene led to pronounced alterations of CK metabolism. By contrast, multiple knockout lacking all *AtLOGs* but *AtLOG*6, *AtLOG*7, and *AtLOG*9 (the authors of the study did not create mutants of *AtLOG*6 and *AtLOG*9) had CK levels comparable to WT plants [145]. Remarkably, *At*LOG7 is the only protein with slightly higher activity than the rice LOG protein, with the rest of the proteins having much lower activity. Additionally, *At*LOG7 has the most consistent activity towards all isoprenoid CKs, but its expression is rather low [25]. Following an opposite pattern, the most highly expressed gene is *AtLOG*8, but the corresponding enzyme has the lowest activity [25]. Thus, *At*LOG7 and *At*LOG8 are other examples that follow the criteria of Hulk and Deadpool, respectively. 

In exceptional cases, enzymes of CK metabolism can occur as chimeras. A fusion protein of isopentenyltransferase and Lonely guy was discovered in *Rhodococcus fascians* [146], *Claviceps purpurea* [137], *Fusarium pseudograminearum* [82], and several other *Fusarium* species [147]. Most species of the *Fusarium fujikuroi* species complex have two genes coding for such a fused protein. *Fusarium proliferatum* ET1 has even three *IPT-LOG* genes [147]. Furthermore, all of the fungal fusion genes are clustered with a cytochrome P450. *Cp*IPT-LOG was able to perform both activities and it showed higher preference for DMAPP over HMBDP. Notably, ADP and ATP did not serve as substrates [137].

These organisms with IPT-LOG fusion proteins are all pathogens that utilise Hulk CKs to inundate their host with active CKs to override the tightly balanced production of CKs normally required by plants. Thus, it is advantageous for the pathogens to link the production of CKs with their activation, which is unlike the need for plants to more strictly regulate the spatio-temporal distribution of both activities. 

Although LOG may provide the most straightforward option for CK activation, there is some circumstantial evidence indicating there is a significant occurrence of the two-step activation. First, there are considerable amounts of ribosides *in planta*, often at similar, if not higher, levels than those of the nucleotides (e.g., [19,49,81]). If LOG activation was the main option, the ribosides would be expected to be present in only minute quantities and the nucleotides would be the major forms among the active trinity of nucleobases, nucleosides, and nucleotides. Second, the ribosides are considered to be the main transport forms of CKs [148,149]. The CKs could be dephosphorylated before xylem loading and phosphorylated upon unloading from xylem sap by nonspecific phosphatases/kinases. This would ensure an effective distinction between the translocation and local forms of CKs, which is analogous to sugar transport via sucrose as the transport form. Considering that grafted WT roots are not able to rescue the phenotypes of *log* shoots [113], it appears that, if riboside is truly the main translocation form, indeed, the unloading must be coupled to phosphorylation followed by dephosphoribosylation at the place of action. Further, Osugi et al. [149] ascribe the restoration of *t*Z in the shoots of *log*/WT grafted plants to direct root-to-shoot *t*Z transport. Alternatively, it could be activated, at least partly, through the two-step process upon *t*ZR transport. Third, knocking-out *LOG* genes does not diminish the levels of CK free bases (e.g., [145,150,151]). On the contrary, plants with altered expression levels of the enzymes from the two-step pathway have altered levels of CKs [152,153]. Lastly, CKs fed to plants are often ribosylated and/or phosphorylated [34,73,145]. Thus, at least in some instances, the CKs are interconverted by way of the nucleoside form. 

Although the enzymes of the two-step activation pathway often prefer adenine-type molecules as substrates, there are certain exceptions [154,155]. Moreover, while LOGs have a *K_M_* higher than 5 µM [25,136], the enzymes of the two-step activation pathway may prefer Ade-type compounds, but their *K_M_* for CKs is typically lower than 5 µM (e.g., [154,155,156,157]). This would mean that these enzymes are quickly saturated and work slowly but steadily providing a Deadpool signal. On the other hand, the LOGs are more responsive to the changes of CK concentrations and thus may better respond to the surge of Hulk signal.

Considering that all of the enzymes of the salvage pathway may operate in both directions if provided with appropriate substrates, there is another possibility. The enzymes may in fact operate in the direction of free base -> riboside -> ribotide and serve as a deactivation pathway, as has been shown for the CK riboside phosphorylase [155].

Nevertheless, even when LOG does activate the CKs, hydrolysis of β- and γ-phosphates must occur first, as ADP and ATP are the preferred substrates of plant adenylate ITPs (e.g., [158,159]). It was reported that in some organisms, *LOGs* are organised in operons or as fusions with certain phosphoanhydrases or Nudix hydrolases [138]. Proteins coded by these genes could be responsible for the hydrolysis of CK nucleotides.

## 6. Deactivation of Cytokinins

There are several options for CK deactivation, when they are present at higher quantities than needed. The first option is ribosylation, which can be followed by phosphorylation [34,73,145]. The other option is glycosylation at the side chain of zeatins and topolins or on the adenine ring. The final known option is irreversible degradation by CKX. Alternatively, they may be transported to another cell compartment.

While conjugation of auxin, jasmonic acid, and salicylic acid with amino acids is well established, the formation of CK conjugates with amino acids is an unspoken secret in the CK community. However, unlike the other phytohormones that form expectable amides, CKs bind alanine and homoalanine unusually through the terminal carbon of their side chain to the adenine’s endocyclic nitrogen to form lupinic acid and discadenine, respectively (Figure 4). 

Lupinic acid (and analogous dihydrolupinic acid) was reported in the 1970s and 1980s as a product of (dihydro)zeatin (riboside) feeding to explants or calli of the *Fabaceae* family (e.g., [160,161,162]) or to immature lupin or apple seeds [163]. An analogous compound was formed when BAP was fed to the plants [164,165] and the enzyme used a wide range of compounds with iP being the best substrate [166]. However, it has not been reported since that time and, thus, it is not clear how widespread this enzymatic activity is, the importance of this pathway, and whether there are other analogous compounds *in planta*. 

On the other hand, discadenine was identified as a sporulation inhibitor in the amoeba *Dictyostelium discoideum*. It was shown to exert CK activity in tobacco callus bioassay and modest activity also in *Amaranthus* bioassay [167,168], but it did not affect the growth of *Dictyostelium* [169]. 

Possibly, *Claviceps purpurea* may also utilise a similarly specialised compound as the mutant in the *CYP450* gene showed much higher sporulation rates [137]. Although both organisms belong to different supergroups [170], it would be possible that they both use a similar mechanism for sporulation regulation involving a CK derivative.

### 6.1. Glycosylation

CKs can be glycosylated either at the nitrogens of the purine ring or at the hydroxyl of the side chain of zeatins or topolins. Many conjugates of CKs have been identified in the past. Besides the canonical glucosides and ribosides, there are also *O*-xylosylderivatives of (dihydro)zeatin [171], CK ribosides with a glucose attached to ribose [172,173], or zeatin riboside with a branched 5-sugar chain on the side chain [174]. One could include also *O*-acetyl-(dihydro)zeatin identified in yellow lupine [175], although this is an ester and not a glycoside. Nevertheless, the significance of these modifications is challenging to estimate as there have been no follow-up studies.

Glycosylation of CKs on the side chain leads to a loss of activity, because the modified side chain can no longer be recognised by receptors. The *O*-glucosylation is thought to be reversible, since β-glucosidase from maize is capable of hydrolysing CK *O*-glucosides [176] and the glucosides can thus operate as a storage form [177]. However, it is their subcellular localisation that determines their fate. It has been noted for decades that plants accumulate *O*-glucosides in mature or senescing leaves [178]. As exogenous *O*-glucosides possess anti-senescence activity (e.g., [179]), a strict compartmentation in plant cells was proposed. The *O*-glucosides were shown to be present predominantly in the apoplast and to a lesser extent in vacuoles [180,181]. Additionally, they were reported in chloroplasts after prolonged darkness [182,183]. In the apoplast, they probably originate from deactivation to prevent further activation of the receptors, and in vacuoles, they are stored until being degraded or re-used. Possibly, vacuolar CKXs physically interact with yet unknown glucosidases to prevent leakage of CKs to cytosol during degradation of CK *O*-glucosides. It is noteworthy though that the majority of the *O*-glucosides is in the apoplast [181].

Targeting maize β-glucosidase to vacuoles led to decreased accumulation of *t*ZOG in tobacco grown on medium with *t*Z, while localisation to chloroplast (its natural compartment) increased it [184]. Hence, *O*-glucosides localised in vacuole are usually not hydrolysed, and targeting β-glucosidase to this compartment depletes these pools, and other means of deactivation must be found. On the other hand, localising it to plastids leads to increased turnover of CKs and their release into other compartments until they are terminally inactivated in a vacuole.

It has been suggested that the *O*-glucosides are transported both into chloroplasts and vacuoles as conjugates [184]. However, a more parsimonious explanation would hold that CKs are synthesised in the chloroplast [23], where the *O*-glucosylation occurs at night [182] when active CKs are not required. Hence, the CK *O*-glucosides serve as storage forms in the chloroplast. When the CKs are required, they are hydrolysed by a β-glucosidase and transported into the subcellular compartment as needed. Once no longer required, they are glucosylated and transported into the vacuole, where they are stored as inactivation products. Whether they are first glucosylated and then transported into the vacuole or vice versa is currently not possible to confirm, but there are no known transporters of CK glucosides. On the other hand, no known CK glucosylating enzymes were reported to localise to the vacuole either. However, neither were they reported to localise to chloroplasts where the *O*-glucosylation occurs.

A modification of the adenine ring diminishes binding to CK receptors [58], but enzymes of CK metabolism, notably the catabolic CKX [18,19], and possibly also the trans-porters, may still recognise these metabolites. Thus, the *N*-glucosides are considered the deactivation products. However, the ability to *N*-glucosylate CKs was clearly gained later during evolution as cyanobacteria, algae, and bryophytes generally contain low to null amounts of CK *N*-glucosides (cf. [27,51,52]). The question, however, remains: why have plants evolved another system for CK deactivation if it were terminal, when another system works perfectly well? The glucosylation is rather a Deadpool form of inactivation as it operates already at low disturbances of CK levels, when the Hulk deactivation by CKX is not necessary (see, e.g., [21]). Intriguingly, the ability to glucosylate CKs may have evolved to eliminate the need for degradation by CKX, which produces toxic aldehyde, as has been observed for *Mycobacterium* [138].

Cytokinin *N-*glucosides were sometimes reported as active CKs at the beginnings of CK research (e.g., [179,185,186]. In some cases, their activity was even higher than that of the corresponding free base. However, once they have been shown not to bind to CK receptors [58], they were re-branded as inactive compounds. However, recent research suggests that *N*-glucosides indeed show activity in certain bioassays [17,187]. The activity seems to be limited to anti-senescence effects, though (Figure 5). It is actually known for a long time that structural requirements to affect processes not involving growth are laxer [7]. Moreover, targeting the CKX into the vacuole, which contains predominantly *N*-glucosides [181], leads to a typical CK-deficient phenotype [188,189]. This suggests that vacuole-stored CKs are still important for the plant.

There are several possible modes of action of CK *N*-glucosides:
I.They are active *per se* through:Binding to CK receptors—currently, *N*-glucosides were shown not to bind to CK receptors, but the reported specificities may not reflect situations *in planta* as could be illustrated by potent CKs DHZ or BAP, which are often reported with dissociation constants one-to-two orders of magnitude higher than *t*Z or iP (e.g., [13,58,68,70,190]).Binding to other receptors or binding proteins—CKs and their derivatives are known to bind various proteins, including cyclin-dependent kinases or CK binding protein [191].Alteration of CK metabolism, e.g., by inhibition of CKX or conjugation enzymes, which would prevent CK inactivation and thus increase the concentration of active CKs. This mode of action could work also in the case of the abovementioned BAP and DHZ to explain the discrepancy with the affinity of CK receptors and their biological activity.II.Hydrolysis to free bases, which in turn exert the activity; however, this would not explain the differential regulation of gene expression upon treatment with free base, *N7*- or *N9*-glucoside [187]

The *N*-glucosides are usually referred to as resistant to hydrolysis, but there are several reports of their hydrolysis. For example, kinetin *N3*-glucoside is cleaved by β-glucosidase (other *N3*-glucosides were not tested; [176]) and activity of DHZ3G in bioassays is also attributed to hydrolysis [192]. Further, tobacco cells converted BAP7G to a nucleotide, hence hydrolysis of the glucose had to have occurred [193]. Similarly, WT tobacco leaf discs hydrolysed BAP3G and BAP9G but not DHZ7G or DHZ9G [194]. Additionally, free BAP is released in soybean from its *N9*-tetrahydropyrrol and *N9*-tetrahydrofuran derivatives [195] and *N9*-tetrahydropyrrol and *N9*-glucosides of 3-methoxyBAP, BAP, and DHZ are hydrolysed in maize but not in *Arabidopsis* [75]. Recently, *t*Z7G and *t*Z9G were shown to be hydrolysed in *Arabidopsis* seedlings and a cell line, while no hydrolysis of iP7G, iP9G, or *c*Z9G was observed [196]. However, concentrations of most metabolites remained about the same for the first 100 min of treatment. Hydrolysis of *t*Z9G in oat leaf segments or in excised *Arabidopsis* cotyledons was not observed (Hluska, unpublished results). The notion of resistance of *N*-glucosides to hydrolysis comes from work on a sole enzyme [176], but it is possible that other hydrolases exist that can release active CKs from their conjugates.

The first described enzyme glucosylating CKs was a cytokinin 7-glucosyltransferase from radish cotyledons [197]. It could use a wide range of substances, including adenine or aromatic CKs. Among them, compounds with a saturated side chain were preferred.

An interesting comparison for the study of CK glycosylation came from the common bean (*Phaseolus vulgaris*) and lima bean (*Phaseolus lunatus*). These species are related, yet they differ in their ability to glycosylate CKs and possibly also to degrade them (as hypothesised by Mok et al. [30]). The xylosyltransferase (*t*ZOXT) from *Phaseolus vulgaris* uses exclusively UDP-xylose, while the glucosyltransferase (*t*ZOGT) from *Phaseolus lunatus* is able to use both UDP-glucose and UDP-xylose [171,198]. Meanwhile, the *t*ZOGT was specific for *t*Z, while the *t*ZOXT could utilise both *t*Z and DHZ. Neither *c*Z, nor *t*ZR were substrates for either of the enzymes. The genes were later identified [199,200].

While both cytokinin *O*-glucosyltransferases identified in maize have strict specificity for *cis*-zeatin and were thus designated as *cis*-zeatin *O*-glucosyltransferases (*c*ZOGT; [42,57]) with only weak activity towards *t*Z [42], there are five CK-specific glucosyltransferases in *Arabidopsis* [201] and they do not strictly discriminate between the zeatin isomers. Of these, two produced *N*-glucosides with preference for *N7*-glucoside, and showed no discrimination of substrates with the exception of BAP. The other three enzymes produced *O*-glucosides, but two of them had very low activity. They were reported later to utilise other substrates [202,203]. However, UGT85A1 glucosylates *t*Z and *c*Z with similar activity, while DHZ was the least effective substrate.

It should be noted though that the five *Arabidopsis* enzymes were identified in a screen of 105 glucosyltransferases for their ability to *glucosylate* five CKs, which did not include *cis*-zeatin. Thus, it is possible that other glycosyltransferases, using other sugar donors and/or *cis*-zeatin as an acceptor, exist in *Arabidopsis*. Furthermore, there was no genome or “glucosyltransferase-ome”-wide search for CK-specific enzymes in maize. Therefore, it is still possible other CK-specific glycosyltransferases are present in maize.

With all of this considered, the differences in preference of *O-*glucosyltransferases in these model organisms for *c*Z-type- and *t*Z-type-dominant species led to the proposal of a novel hypothesis explaining the prevalence of isomers of zeatins in each of them [81]. Rather than a selective drive towards the preference for one isomer over the other, the substrate specificity of the *O-*glucosyltransferases seems to have resulted in a random drift towards a predominance of one isomer or the other. A kingdom-wide analysis of CKs [27] showed that preference for one zeatin isomer or another is not restricted to a particular taxon but is in fact a random occurrence. This is compatible with a single locus being solely responsible for this divergence rather than a complex network that would require more mutations to switch from one isomer system to the other. Furthermore, expression of *t*ZOGT in maize was observed to lead to equal amounts of *c*Z-type and *t*Z-type CKs in mature plants [204], thereby confirming, once again, that the selectivity of *O-*glucosyltransferases may be responsible for the prevalence of either one of the isomers.

The existence of *c*Z-type-dominant plants would suggest that *c*Z may function as Hulk CK in certain species, especially as it exerts a similar activity in bioassays employing such plants [38,46]. However, this hypothesis suggests that even these plants are not capable of fast and inducible *c*Z production and thus it probably acts as a Deadpool even in these plants. Nevertheless, more testing would be required to dissect this question.

Recently, two CK-specific glucosyltransferases putatively forming *O*-glucosides were suggested to be involved in ovule development through regulation by CUP-SHAPED COTYLEDON1 (CUC1) and 2 [205]. However, the authors identified UGT85A3 and UGT73C1 as the enzymes linking CUC transcription factors and CK metabolism, although the first one was shown to lack activity towards CKs [201] and the second exerts much higher activity towards TNT derivatives [202]. Further, the authors reported a decrease of CK free bases and an increase of CK glucosides in knockdown plants. Yet, the biggest decrease is observed for iP, which cannot form *O*-glucosides. Additionally, there is a much bigger increase of CK *N*-glucosides in contrast to CK ribosides *O*-glucosides. Last, there was no direct data to indicate that these enzymes indeed glucosylate CKs. Plants with altered levels of these *UGTs* had a similar phenotype as the plants with altered *CUC* expression. This was expectable as the *UGTs* were deregulated in such plants. Nonetheless, the authors have not determined the CK content in plants with altered *UGT* expression or in vitro activity of these enzymes.

Rice and maize, both *c*Z-type-dominant plants, were transformed with *c*ZOGT and *t*ZOGT [38,204,206]. Predictably, the plants had increased levels correspondingly of mainly *c*Z(R)OG and *t*ZOG. Both species had reduced shoot growth (a CK-deficient phenotype). With the exception of Shang et al., the authors unexpectedly also observed elevated levels of chlorophyll and delayed senescence, which are traits usually associated with high CK levels [207]. Indeed, the maize plants showed increased levels of *t*Z and iP in the leaves. Additionally, roots were longer, although rice had a reduced number of crown roots [38,206]. Thus, in general, the phenotypes were similar. The only notable difference was the development of sexual organs and seeds, which was highly affected in the maize expressing *t*ZOGT [204]. Mainly the size of tassels was reduced by 75% in heterozygous plants and they were feminised to varying degrees in homozygous plants [204].

### 6.2. Cytokinin Oxidase/Dehydrogenase

The enzyme that degrades CKs is one of the longest studied of those involved in CK metabolism and it is, thus, the best characterised one. The topic of CK degradation has been reviewed extensively by others [208,209].

The enzyme was first described in the 1970s [210] and cloned independently by two groups almost 30 years later [211,212]. It is a flavoenzyme with covalently bound FAD using artificial electron acceptors [213]. Putative natural electron acceptors were identified [214]. Because the enzymes efficiently use Q_0_, an isoprenoid side chain-deprived analogue of ubiquinone, it was proposed that the cytokinin oxidase/dehydrogenase enzymes may be associated with membranes [213]. Recently, *At*CKX1 was shown to be a transmembrane protein localised to the endoplasmic reticulum (ER; [215]).

*CKXs* form small gene families. The respective enzymes differ in subcellular localisation, temporal and spatial expression, and substrate preferences. The most studied are enzymes from *Arabidopsis* and maize. Several genes were cloned and heterologously expressed [132,211,212,216,217,218]. Characterisation of the complete family of CKX enzymes was performed in *Arabidopsis* and maize [18,19].

The CKX expression and/or activity is usually induced upon CK application [219,220,221,222]. It was suggested that CKX may play a role of a “detoxifier” [221]. However, in few exceptions, higher CK content led to decreased expression of (some) *CKX* genes (e.g., [21,43,223,224]). This may result from various needs for particular isoforms due to their substrate specificities and/or subcellular compartmentalisation.

CKX is able to efficiently degrade all types of natural CKs, with the exception of resistant DHZ [18,19] or aromatic CKs that are very weak substrates [18,218]. Additionally, *c*Z was considered as a weak substrate, but there are several enzymes that cleave it preferentially. Among them, *Zm*CKX8, 9 and 10 [19] or *At*CKX1 and 7 [27]. Indeed, *Arabidopsis* overexpressing *At*CKX1 or *At*CKX7 had drastically reduced levels of *c*Z-type CKs [55]. Considering that AtCKX7 and *Zm*CKX10 are the only CKX from the respective organisms localised to cytosol, the preference for *c*Z-type CKs may not be a coincidence as they may be involved in removal of the cytosol-localised Deadpool tRNA-derived *c*Z.

The effect of CKX on important agricultural traits has been known at least since the iconic identification of *OsCKX*2 as the determinant of grain yields in rice [225]. Similar results were observed in other works [226,227,228,229] for which lower *CKX* expression led to more tillering, an increased number of siliques and/or seeds, and thus to higher yield. Interestingly, Zhang et al. [230] observed a positive correlation between the grain number per spike and the expression of two *TaCKX*2 genes in young spikes. These genes are the most similar to *Os*CKX2 and *At*CKX3 described in the other studies, the expression of which correlates negatively with these agricultural traits.

Constitutive expression of *CKX* leads to stunted shoot as was shown in the influential work by Werner et al. [56] as well as in other works (e.g., [231]). Thus, the reduction of CK levels by means of *CKX* root-specific overexpression became a well-defined approach to crop improvement. It leads to increased root systems both in the model plants *Arabidopsis* and tobacco [232] as well as in agricultural plants, such as barley [189,233], rice [234], and poplar [235]. This leads to better dealing with drought stress [189,232,233] and/or improved mineral uptake [232,233,236].

An interesting tactic would be to combine both approaches, i.e., upregulation of *CKX* expression in the roots and downregulation of the *CKX* expression in the shoots and/or grains, and observe whether this leads to synergistic strengthening of the sink in the seeds and further enlargement of yield, possibly also with improved mineral content as has been observed for other combinations of traits [237].

The involvement of CKX influence on agricultural traits was recently reviewed in more depth in an excellent review by Chen et al. [238].

## 7. Cytokinin Signalling

The transduction of a CK signal relies on a modular system similar to that used by bacteria known as two-component signalling, which reflects that it is composed of hybrid histidine kinase (HK or CHK for CHASE-domain containing histidine kinase; see Heyl et al. [239]) receptor and response regulator (RR). In plants, an additional component is required—a histidine-containing phosphotransfer protein (HPT)—for transfer of the signal to the nucleus (Figure 6). The CK signal circuitry was reviewed in more detail recently [12,240].

There are three CK receptors in *Arabidopsis*. The CK receptors contain a CHASE domain at the N-terminus surrounded by two or more transmembrane helices. The CHASE domain was reported to contain two PAS(-like) domains [241,242]. However, the true PAS domains are present only in the cytosol and the extracytosolic PAS domains were assigned to a distinct Cache superfamily [243]. Thus, the CK receptors contain CHASE domains (without any subdomains) newly assigned to the Cache superfamily. The recently described cytokinin-sensing histidine kinase from *Xanthomonas* also contains two true intracytosolic PAS domains in addition to the CHASE domain [244].

Among the *Arabidopsis* CK receptors, AHK4 is unique: it is highly specific for free bases of *t*Z and iP [58] and it possesses phosphatase activity in the. absence of CKs [245]. The AHK3 and the maize receptors also generally recognise *c*Z, and DHZ (and their respective ribosides) with affinities similar to those of other isoprenoid CKs [58,59,69,70]. A somewhat opposite situation was observed in an apple tree for which *Md*CHK2 (a homologue of AHK2) showed the widest substrate acceptance and highest sensitivity at the same time [14]. Interestingly, it is also the most highly expressed gene.

Potato CK receptors also showed partial differences in CK preferences in comparison with AHKs [190]. Notably, the potato receptors did not perceive *c*Z with high activity, although it is present in potato at higher concentrations than the *trans*-isomer [33,34,35,36]. This would add to the abovementioned discrepancy of CK receptors not perceiving what are thought to be active CKs.

There is discrepancy concerning the subcellular localisation of the CK receptors. The first reports stated in 2011 that cytokinin receptors may not be present mainly on the plasma membrane [246,247]. Since this time, there were more reports supporting this opinion as reviewed by the authors of those papers [248], changing the view from endocycling to strictly functioning on ER. A favourite argument is the pH optimum of the receptors in a neutral-to-mildly basic range, which should correspond to ER lumen rather than apoplast (e.g., [13,190,249]). However, the apoplastic pH lies mostly in the range 5.0-6.5 [250], which is the range where the receptors change their activity. Thus, operating in this region provides another level that is subject to regulation. Indeed, the pH of apoplast is known to change in response to various cues [251]. For example, nitrate, known to induce shoot growth through action of the Hulk CKs [24,113,114], increases the apoplastic pH above 6.0 [252], where the receptors reach their maximum binding capacity. Originally, it was also suggested that this responsiveness to pH change may contribute to auxin-CK crosstalk [70] as auxin decreases pH in the apoplast. Recently, it was suggested that the ability of the receptors to form dimers is also pH dependent [249]. However, if we accept the premise that the pH of the apoplast is changing, that would also be irrelevant. Further, it is AHK3 that responds strongly to pH change [13] and it is localised to a lesser extent on the plasma membrane [247,253] as compared to AHK4, which does not respond so strongly to the pH change. The poplar homologues of AHK4 showed, in fact, a negative correlation between *t*Z binding and pH [254].

Moreover, recent experiments confirmed that functional receptors are localised on the plasma membrane [253,255]. While Antoniadi et al. [253] showed that extracellular CKs are capable of CK receptor activation, Zürcher et al. [255] showed the significance of CK removal from the apoplast for proper development. Indeed, there are several apoplast-localised CKX enzymes (e.g., [132,256]). However, this does not preclude additional localisation of the receptors to the ER. Expression of an ER-targeted anti-CK antibody led to developmental abnormalities [257]. Similarly, the *At*CKX1 was shown to localise to the ER [215]. A higher proportion of AHK4 was reported to localise to non-ER membranes though [247,253]. Interestingly, perception of the extracellular CKs was most impaired in plants lacking AHK4, even in a single mutant [253]. Moreover, Kubiasová et al. [258] showed that the subcellular distribution of AHK4 varies in different cell types with actively dividing cells containing AHK4 on the plasma membrane. Further, CKs were shown to regulate endocytosis of several proteins [259,260]. This would be an additional negative feedback loop in CK signalling by the removal of the receptors. Thus, the AHK4 on the plasma membrane, recognising mainly *t*Z and iP free bases, may perceive long-range Hulk signals in the vasculature with strong effects on the overall plant architecture. By contrast, the majority of AHK2 and AHK3 on the inner membranes perceive a wider range of CKs, including *c*Z, originating from tRNA hydrolysis, serving also as the local autocrine Deadpool signal.

In conclusion, there are *in planta* experiments showing localisation of functional receptors on the plasma membrane. Although the experiments of Zürcher et al. [255] could not have differentiated between CK receptors functioning on the plasma membrane and those working on the endomembranes, if endocytosis was operational, it was shown that only a negligible fraction of the immobilised CKs goes inside cells [253]. Moreover, the point made about the receptors’ binding capacity responding to pH is in fact an argument for the receptors’ operation on the plasma membrane. However, as has been mentioned above, regulation of the CK levels in the ER is also crucial for proper plant development. Thus, likely, in the spirit of the original reports [246,247], CK receptors are functional on both membranes.

Another matter of dispute pertains to the ability of CK receptors to bind CK ribosides. Those were traditionally grouped with free bases, and classified as active forms because of their activities in various bioassays. Indeed, the original measurements with CK receptors in the heterologous *E. coli* system confirmed their binding [58,59]. It was suggested later that only eukaryotic membranes should be used to evaluate CK receptors’ binding properties [13]. The authors concluded that ribosides are not bound by the receptors. Although not entirely clear from the presentation, the data seem to suggest that at least *Zm*HK1 binds iPR with a dissociation constant one and two orders of magnitude larger than *t*Z and *c*Z or iP, respectively. This is about three times lower than the dissociation constant for the binding of DHZ. However, the authors did not measure activation of the signalling cascade, which may have been prevented by sterical hindrance. Nevertheless, Daudu et al. observed the growth of yeast expressing apple tree *Md*CHK2 on CK ribosides at concentrations at least 100-times higher than of free bases [14]. Notably, those authors also observed growth on *t*ZOG at the same concentration as on iPR.

Upon CK binding, the receptors putatively form dimers and transfer phosphate to HPTs. Such dimerisation was observed several times (e.g., [14,246,247,261,262]). Noteworthy, studies that used BiFC did not include proper negative controls [263]. Indeed, such studies detected all tested interactions. In other instances, often heterodimerisation is observed, while formation of homodimers is not. Yet, the heterodimerisation does not seem to be physiologically relevant [264]. This begs the question of whether the observed interactions reflect true contact or an artefact. Moreover, the putative inability to form homodimers contradicts the ability to detect activation of two-component signalling in heterologous systems where only a single receptor is expressed.

Recently, CK signalling was elucidated in bacteria [244,265]. The receptor from *Xanthomonas campestris* pv. *campestris* specifically binds iP. Rather than activating the phosphorylation cascade, this binding lowers the phosphate load in the system, ultimately leading to hydrolysis of 3’,5’-cyclic diguanylic acid [244]. In the human pathogen *Mycobacterium tuberculosis*, CK induces the formation of a lipid-like molecule, which binds to a repressor and de-represses the expression of a responsive gene. However, unlike canonical CK signalling, the *Mycobacterium* one is significantly inhibited by adenine [265].

Following phosphorylation, HPTs transfer the phosphate to downstream proteins in the nucleus [266]. However, it was shown that the localisation of HPTs is independent of CK perception [267]. Unlike the other HPTs, the *Arabidopsis* HPT6 has the conserved Asp replaced with Asn and thus cannot accept a phosphate. It competes with the other HPTs in binding to the receptors and thus attenuates the CK signalling [268].

Among the receiver proteins are Response Regulators, which are classified as type-A, -B, and -C, based on their domain structure and responsiveness of their expression to CK treatment [269]. Type-B RRs are positive regulators of the CK response and they serve as transcription factors directly influencing the expression of response genes [270]. Additionally, *type-A RRs* belong to the response genes [271] and serve as negative regulators of the CKs, probably by competing with type-B RRs for the phosphate. Thus, they serve as a negative feedback loop. The function of type-C RRs is less thoroughly characterised [12]. Their structure resembles that of type-A RRs, but on the amino acid level, they are more similar to the receiver domain of the receptors and their expression is not upregulated by CKs [272]. Spatially, their expression is restricted to reproductive organs [272,273].

Cytokinin Response Factors (CRFs) are land plant-specific members of the AP2/ERF protein family [274,275]. They are characterised by the CRF domain, single AP2/ERF domain, and CRF1-6 also contain the MAPK phosphorylation site motif [275]. CRF proteins form homo- and heterodimers and interact also with HPTs and rarely with response regulators. The CRF domain is solely responsible for the interaction [276]. However, only some of them are upregulated by CKs [277]. The CRFs were implicated in response to a range of abiotic stresses [278,279,280,281,282,283,284].

Lastly, GL1 ENHANCER BINDING PROTEIN (GeBP) and GeBP-like (GPL) transcription factors were implicated in CK signalling [285]. These unconventional leucine zippers form homo- and heterodimers, which localise to the nucleus. Triple mutant *gebp gpl*1 2 did not arrest its growth in the presence of kinetin unlike WT. In general, this mutant phenocopied the double mutant of type-B RRs. However, no further studies in relation to CKs have been reported.

## 8. Cytokinin Transport

Over the years, there have been several indirect lines of evidence for CK transport. CKs serve as a long-range signal for nitrogen [24,286], and *t*Z- and iP-type have been detected in xylem and phloem sap, respectively [40,148,286,287,288,289]; experiments with grafted plants have repeatedly shown the ability of WT roots or shoots to complement for a detrimental mutation [120,290,291] and overall, the transport of CKs has been shown [175,178,292]. Lastly, the expression sites of *IPT* and *LOG* genes do not overlap with the CK response domains [12]. However, for a long time, the CK transporters had remained elusive.

The first transporter family implied in CK transport were purine permeases (PUPs), with *Arabidopsis thaliana* purine permease 1 (*At*PUP1) inhibited by kinetin and zeatin and to a lesser extent by their ribosides [293]. Later, direct transport of *t*Z was shown [294]. Two rice orthologues were also implicated in grain development due to CK transport [295].

Other candidates for CK transport are the equilibrative nucleoside transporters (ENTs), which transport purine and pyrimidine nucleosides. As the name suggests, in principle, they are bidirectional, and the compound concentration determines the direction. However, circumstantially, the adenosine uptake is probably coupled with phosphorylation to trap the nucleoside intracellularly [296]. Several members of this family have been shown to be inhibited and/or to transport various CKs [148,297,298]. However, multiple members of the transporter family from *Arabidopsis* were previously characterised and reported not to transport kinetin, zeatin, or *t*ZR [299]. The discrepancy could be due to the use of *t*Z-type CKs, which have much higher *K_M_*, but Hirose et al. [148] showed a reduction of the adenosine uptake by 20–30% even with *t*ZR.

Nevertheless, because of the low affinities towards CKs and comparable or higher affinity of these transporters towards canonical nucleobases or nucleosides, their involvement in CK transport was often disputed [208,300]. Since then, a growing body of evidence has been showing direct involvement of various transporters with CKs *in planta*. By the means of a screen for the suppressor of *IPT* overexpression, *At*ENT8 was identified as a potential CK transporter [298].

Several members of the *ATP-binding cassette transporter subfamily G*, namely *Arabidopsis*
*At*ABCG14 [301,302], rice *Os*ABCG18 [303], and apple tree *Md*ABCG28 and 70 [304], were recently shown to transport CKs. The first indicia was co-expression of *AtABC*G14 with *AtIPTs* and induction of its expression upon CK treatment [302]. All of them are mostly expressed in the roots and in the vasculature in general and their expression is upregulated by CKs. The transporters localise to the plasma membrane [301,302,303]. The knockout mutant phenotypes resembled other CK-deficient mutants [301,302,303]. Similarly, the expression of *MdABC*G28 was decreased in a dwarf line of apple tree [304]. The dwarf phenotype of shoots could be rescued by application of *t*Z but not iP [302,303], while root elongation was less sensitive to *t*Z application in the mutants than in WT plants [301]. The CK content in the mutant roots was higher, while it was lower in the shoots. Indeed, the expression of the response regulator genes corresponded accordingly [301,302]. Grafting experiments showed unequivocally that root-derived *t*Z is crucial for the development of shoots but that shoots do not affect the growth of roots [302]. Later, *At*ABCG14 was implicated in long-range nitrogen signalling, confirming its role in CK root-to-shoot transport [305].

Recently, three non-intrinsic ABC proteins *At*ABCI19, 20, and 21 were suggested to negatively regulate CK responses [306]. The proteins lack a transmembrane domain and are part of a large protein complex localised to ER. It was hypothesised that they work either as transporters of CKs on the ER or they regulate translocation of CK receptors from the plasma membrane as they interact with COPI responsible for retrograde transport [306].

*At*PUP14 was shown to be a negative regulator of CK signalling [255]. The cotyledons of *Arabidopsis* heart-staged embryo do not transmit the CK signal, even though *AHK*4 is expressed there and the downstream signalling pathway is functional as shown by expression of *CKI*1—a constitutively signalling HK. Interestingly, *AtPUP*14 was also expressed in the lateral root primordia, the formation of which is strongly inhibited by CKs [307]. Transport of *t*Z by *At*PUP14 was directly shown, and it was ATP dependent. Transport of radioactive *t*Z was inhibited by free bases of other CKs but not by *t*ZR. Further, CK signalling was diminished in mesophyll protoplast cells upon *At*PUP14 expression or upon the presence of CKX enzyme in the “apoplast” but not in the cytosol of the cells [255]. This clearly illustrates the importance of matching the subcellular localisation of CKs and their receptors for triggering of the signals. Analogous results were observed in barley [189]. Localisation of *At*CKX1 to the apoplast was detrimental to plant development, while plants with *At*CKX1 directed to vacuoles showed only a weak phenotype and any phenotype of plants with cytosolic *At*CKX1 was miniscule.

Hence, *At*PUP14 shall work as a stop signal of the CK signalling by depleting apoplastic CKs and thus inhibiting perception by plasma membrane-localised receptors [255]. Even though the kinetic parameters of *At*PUP14 were not determined, *K_M_* of the homologues is in the range of tens µM [293,294], while the *K_D_* of the receptors is in the range of nM (e.g., [68,69]). However, the same works also showed a strong dependence of PUP activity on pH with a maximum in the acidic range while null activity in the neutral-to-weakly-basic range, i.e., complementary to the receptors. Thus, PUPs are likely another level of regulation ensuring that CK signal is not perceived where it is not desired.

Recently, a novel transporter family was implicated in CK transport. The *Arabidopsis* genome contains two genes encoding Aza-guanine resistant transporters (*At*AZG). Both of them were shown to transport adenine and *t*Z with high affinity [308,309]. Although, neither of the loss-of-function mutants, *atazg*1 or *atazg*2, were distinguishable from WT plants in their ability to transport both adenine and *t*Z [308,309], the *atazg*1 plants had longer primary roots and a higher density of lateral roots. Moreover, root explants from those mutants required a higher concentration of kinetin for shoot regeneration [309]. The authors also concluded that *At*AZG1 functions as an active transporter while *At*AZG2 acts as a diffusion facilitator [308]. Considering that the proteins are identical almost at the 50% level and share similarity at more than 75% [310], it would seem unlikely they would differ so much in their mode of action. Further work should be more attentive to details like normalisation of pH, monitoring of quantities of expressed protein, usage of natural substrates rather than hypoxanthine, and exploring alternative explanations (e.g., both strains may lose similar amounts of hypoxanthine, but *At*AZG1 may be faster or have higher affinity for its uptake).

*At*AZG2 localises both to the plasma membrane and to the endoplasmic reticulum. It is expressed in cells surrounding lateral root primordia, and upon auxin treatment, its expression increases and is more widespread [308]. This is an interesting complementary expression domain to the one of AtPUP14.

Because the knowledge on particular CK transporter families is still rather spotty, it is currently difficult to judge the framing of particular transporters in the Hulks & Deadpool hypothesis with the exception of ABCGs transporting the long-range Hulk *t*Z. It is questionable whether *At*PUP14 discarding the Hulk *t*Z signal shall be labelled as Hulk because of the strong effect, or Deadpool, because it contradicts the Hulk signal. However, it is probably safe to hypothesise that in the mutational studies, the Hulk transporters with strong effects in mutants were identified. On the other hand, members of the PUP and ENT families that were not studied so far may still transport CKs, but their activity and/or expression may be low and/or widespread and thus of little interest when studying a particular process.

## 9. Conclusions

In this review, we presented a novel hypothesis showing that the Cytokinin Universe is full of both (i) strong Hulks with high CK activity, metabolite conversion, and sensing power that drives fast growth spurts and (ii) immortal Deadpools sustaining basal survival, metabolites, enzymes, and sensing mechanisms with lower activity but a long-lasting effect.

Here, we reviewed the metabolism, signalling, and transport of the plant hormones CK. This demonstrates that the production levels, and different forms of CK can determine which of the two different Hulk/Deadpool strategies take place. It also highlights that beyond the quantities and forms of CKs, the ability to translocate CKs to where they are needed and the type of sensitivity of the signal transduction machinery can directly affect the Hulk/Deadpool tactics. We included some of the old knowledge, which is currently neglected, and would be worthy of further investigation as elucidation of the significance of these metabolites or enzymes may be shifting erudition in the Cytokinin Universe and have potential utilisation, e.g., in agriculture.

We attempted to show that the current CK research mostly focuses on the Hulks, such as in cases of iP and *t*Z, *At*IPT3, LOGs, etc. This is probably simply because it is easier to study the large effects of Hulk knock-down or knock-up rather than subtle changes caused by alterations of the Deadpools, which often require multiple knock-outs to observe any phenotype.

Moreover, the focus of current research on the *t*Z-dominant species, *Arabidopsis*, and the neglect of the more dated, and perhaps forgotten, early research has led to many as-of-yet unanswered vital questions. Examples include the importance of such enzymes as zeatin reductase or zeatin *O-*xylosyltransferase, which without renewed and more detailed study cannot be adequately gauged. The recent re-discovery of “novel” CKs hints that there may be more diversity to CKs and their metabolic pathways than is currently reflected by the emerging research topics. We tried to pin-point such old/new topics to generate more deserved attention.

Of course, not every CK, enzyme, transporter, receptor, etc. may be filed clearly as Hulk or Deadpool. In fact, most of them may lay somewhere in between. However, by pointing out to the obvious Hulks and Deadpools, we have tried to alert that not only are the Hulks important for plant development, but that the Deadpools also hence deserve their fair share of attention.

## Figures and Tables

**Figure 1 biomolecules-11-00209-f001:**
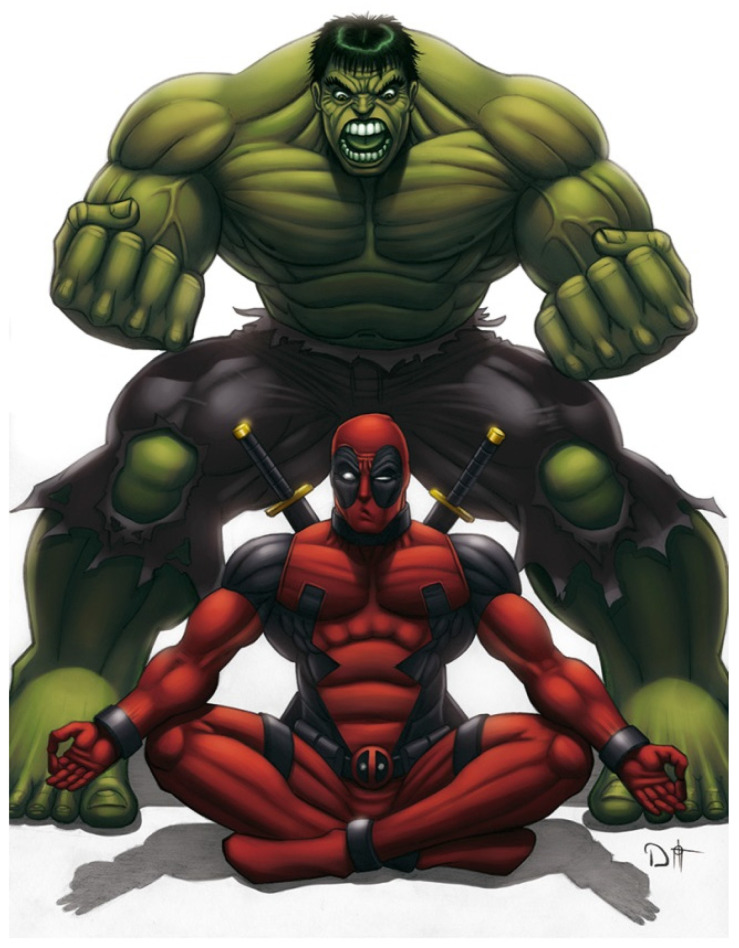
Hulk and Deadpool of the Marvel Universe. Hulk is strong, and uses bursts of brute force. Deadpool on the other hand is resilient due to his immortality. Are there Hulks and Deadpools in the Cytokinin Universe? Drawn by David Ocampo.

**Figure 2 biomolecules-11-00209-f002:**
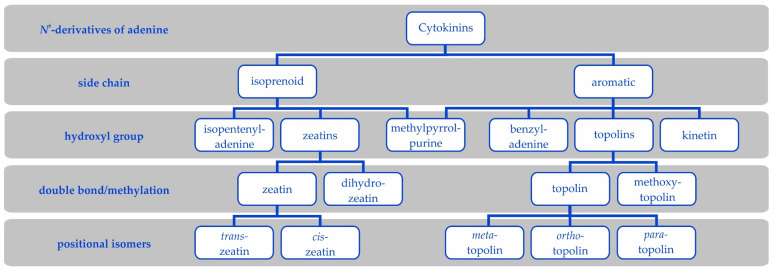
Cytokinin types. Cytokinins are *N^6^*-derivatives of adenine either with isoprenoid or aromatic side chains. A hydroxyl group further differentiates zeatins from isopentenyladenine and topolins from benzyladenine. Further, kinetin is an aromatic cytokinin and 6-(3-methylpyrrol-1-yl)purine is at the borderline of isoprenoid and aromatic cytokinins, because it has an aromatic side chain but originates from *trans*-zeatin. Methoxytopolins exist in analogous isomers as topolins.

**Figure 3 biomolecules-11-00209-f003:**
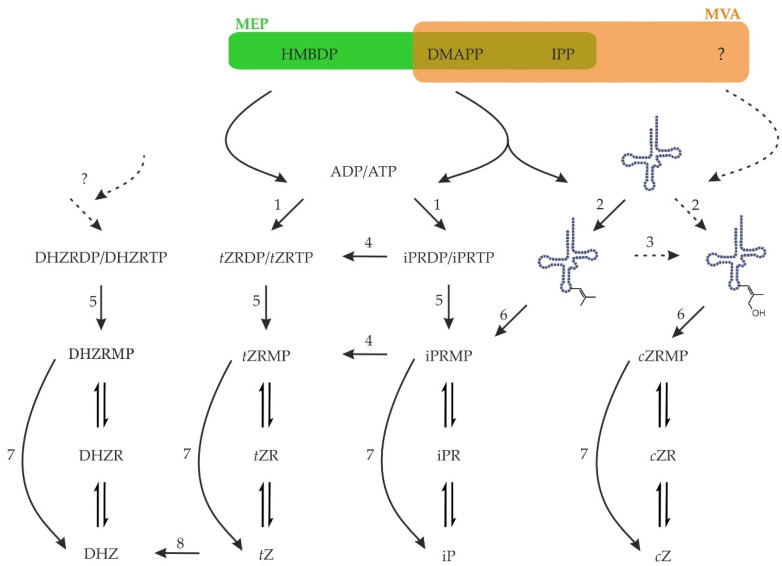
Biosynthesis and activation of cytokinins in plants. Precursors for CK biosynthesis, HMBDP and DMAPP, come either from the methylerythritol (MEP) or mevalonate (MVA) pathways. Plant adenylate IPTs (1) utilise mostly ADP or ATP whereas tRNA IPTs (2) use the adenine in position 37 of certain tRNAs as acceptor substrate. *cis*-Zeatin is known to originate from tRNA, but its synthesis is unclear, as neither a *cis*-hydroxylated precursor nor *cis*-hydroxylase (3) have yet been identified in plants. Nucleotides of iP may be hydroxylated by cytochrome P450 (4) to form *t*Z. Upon hydrolysis of γ- and β-phosphates (5) or tRNA hydrolysis (6), the resulting monophosphates may be activated in one step by CK-specific phosphoribohydrolase named Lonely guy (7). Alternatively, the nucleotides, nucleosides, and nucleobases are probably interconverted by enzymes of purine metabolism. The free base *t*Z may be reduced to DHZ by zeatin reductase (8). Whether there is any *de novo* biosynthesis of DHZ is currently unknown. Zeatin *cis-trans* isomerase does not exist.

**Figure 4 biomolecules-11-00209-f004:**
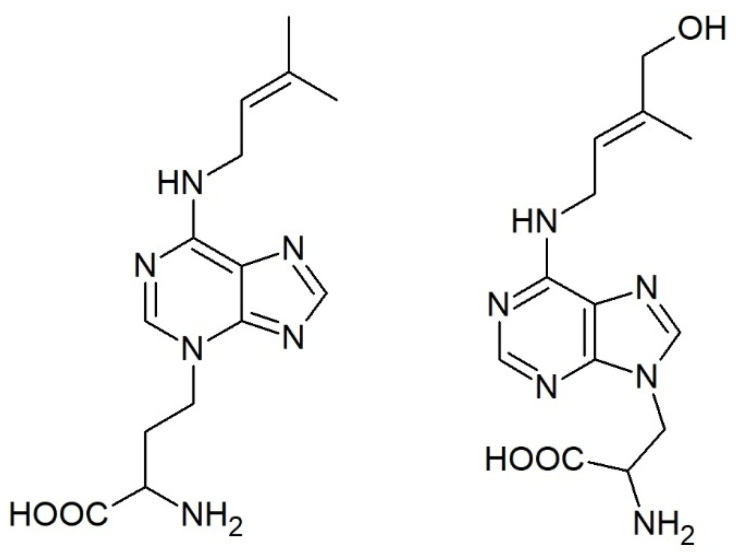
Structures of the only known cytokinin-amino acid conjugates: discadenine (left) and lupinic acid (right).

**Figure 5 biomolecules-11-00209-f005:**
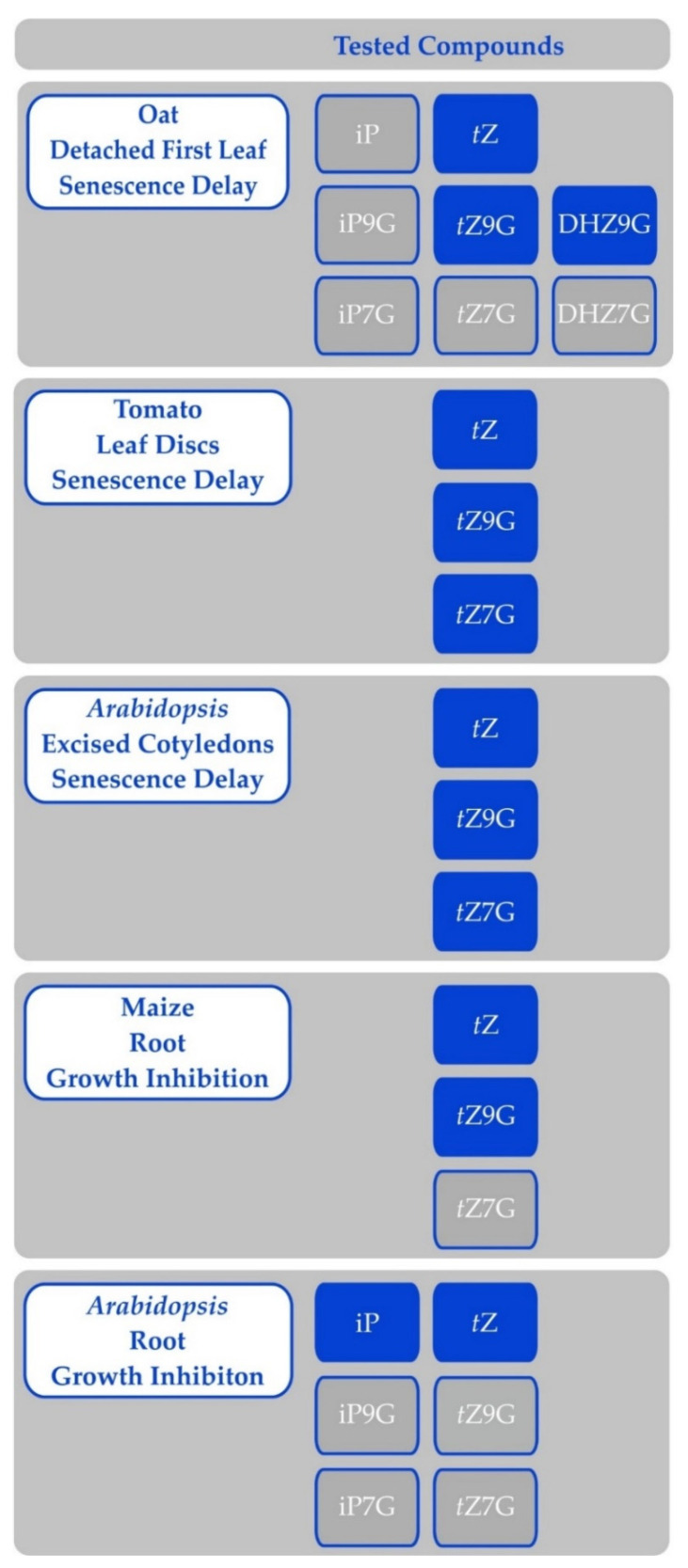
Activity of cytokinin free bases and *N*-glucosides in various bioassays; results of Eva Pokorná and Václav Motyka. Compounds on blue background were active in the specified bioassay, while compounds on grey background were inactive. Any compounds not listed in either column were not tested.

**Figure 6 biomolecules-11-00209-f006:**
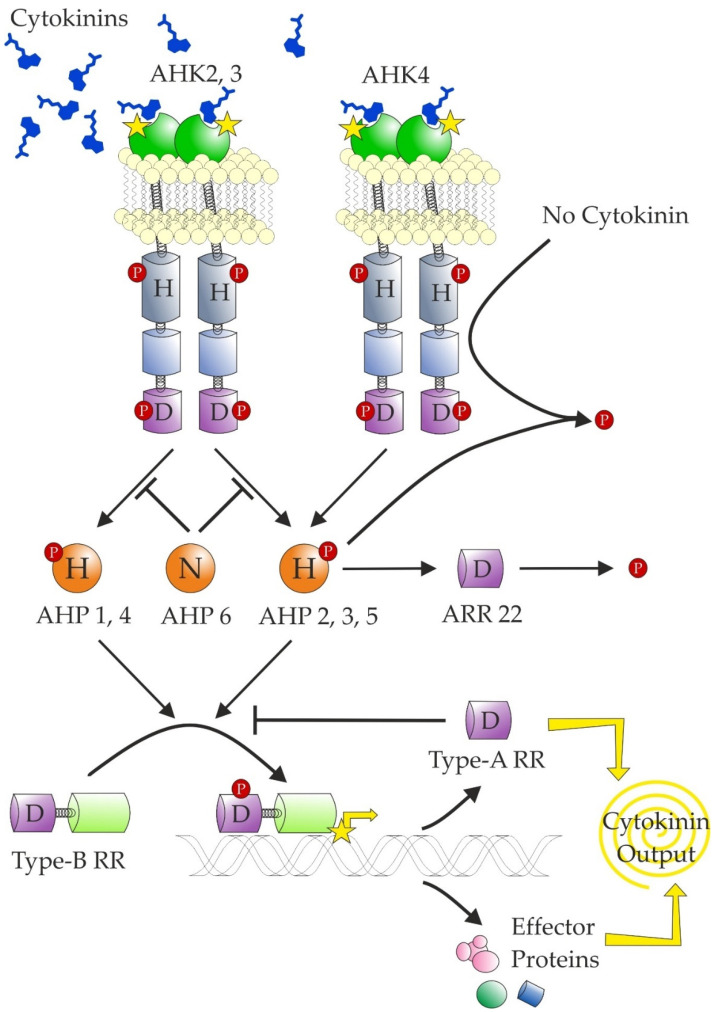
Schematic representation of current knowledge of cytokinin signalling. The transmembrane cytokinin receptors (histidine kinases) auto-phosphorylate in the presence of cytokinins and subsequently transfer the phosphate to the histidine phosphotransfer proteins (AHPs). However, in the absence of cytokinins, AHK4 has phosphatase activity and dephosphorylates all AHPs. The transfer is inhibited by AHP6, which has a substitution of the conserved Asp for Asn. AHP2,3 and 5 are rapidly dephosphorylated by ARR22. The AHPs transfer the phosphate group to type-B response regulators, which work as transcription factors. One of their targets are type-A response regulators, which inhibit the phosphotransfer from AHPs to type-B RRs. The conserved residues are marked (H—histidine; D—aspartate; N—asparagine). The phosphate group is marked by a P contained in a red circle. The extracytosolic CHASE domains are shown in green, kinase domains are shown in grey, and receiver and receiver-like domains are respectively shown in violet (marked with a D) and light blue (no D).

**Table 1 biomolecules-11-00209-t001:** Examples of Hulks and Deadpools.

Hulks	Deadpools	Commentary
*t*Z, iP	*c*Z aromatic cytokinins	While *t*Z and iP induce large changes in plant physiology, *c*Z and aromatic cytokinins are “slow” with weak CK activity, but they are resistant and thus persistent.
*At*IPT3	*At*IPT5, *Sl*IPT3 and *Pp*IPT5a tRNA IPTs	*At*IPT3 is the main enzyme synthesising CKs with spatio-temporally focused production; on the other hand, IPT5 orthologues are expressed widely but they have only low activity, thus producing small less focused amounts of CKs. tRNA IPTs usually exert wide expression and they produce the Deadpool *c*Z.
receptors on PM	receptors on ER	PM-localised receptors may perceive the Hulk CKs *t*Z and iP with very high activity. To diminish Hulk signalling when Hulk CKs are not present, AHK4 also possesses a phosphatase activity. On the other hand, ER-localised receptors perceive autocrine Deadpool signals.
LOGs *At*LOG7	two-step activation *At*LOG8	LOGs activate CKs (almost) in a single step from biosynthesis products specifically in regions of high cell division activities, while the two-step activation functions throughout the plant body at low speed. Among LOGs, one can distinguish *At*LOG7 with high activity and expression focused to high activity hotspots and *At*LOG8 with minimal activity but expression throughout the plant.

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
