# Peer review of "The Hulks and the Deadpools of the Cytokinin Universe: A Dual Strategy for Cytokinin Production, Translocation, and Signal Transduction"

_biomolecules, 2021, doi:10.3390/biom11020209_

Round 1

Reviewer 1 Report

Dear Authors,

I have done the second round of review of the manuscript "The Hulks and Deadpools of the Cytokinin Universe: A dual strategy for cytokinin production, translocation and signal transduction" that has been submitted to the journal "Biomolecules".

In my opinion, the quality of the paper has significantly benefited from the first round of revision and is now closer to its final acceptable form. The main idea of the manuscript represents a valuable, novel concept that deserves to be published but before that there are, in my opinion, more parts that should be left out because in the current form, they still tend to unnecessarily deviate the reader's attention from what is really important in the text.

Although the manuscript has been significantly shortened in accordance with my suggestions upon the first round of review, there still remain lengthy passages that, in my opinion, do not contribute to the clarification of the Hulk vs. Deadpool hypothesis, but instead just represent a burden to the text. These include passages that are not strictly related to the physiological processes that occur in planta, for instance, those that are referring to non-plant organisms (such as, for instance, lines 906-914), or those that are referring to kinetin, but most importantly, lengthy considerations describing the methodological approaches or critical references to the work of other research groups, as is the case with, for instance, lines 1013-1028. I suggest that the manuscript text should be thoroughly revised for passages that refer to non-in planta-processes or molecules not derived from plant tissues, as well as for passages that sound like criticism of other authors work and/or detailed consideration of scenarios that have not been taken into account in particular past publications – and that all such passages should be deleted from the manuscript.

Last but not least, I believe that Table 1 has greatly contributed to the clarification of the Hulk vs. Deadpool paradigm in cytokinin metabolism and signalling by offering clear examples for both phenomena and by illustrating that "Hulk" and "Deadpool" are not two sharply separated compartments in cytokinin metabolism and signalling, but rather two characteristics of the components of the cytokinin machinery that are best defined relative to the other components of the same order (for instance, for AtIPT3 relative to the other IPT enzymes). In my opinion, this Table should be rendered more central to the entire manuscript, in the sense that more solid examples (such as DHZ) can be added to it, but also in the sense that the rest of the manuscript should be mainly focused on "feeding" this table with solid argumentation, while the lengthy parts of text that are completely unrelated to the table can safely be left out... Possibly even by leaving entire chapters out. In this way the authors can obtain a shorter but more focused paper that will be entirely organized around its central idea.

This time I am not giving point-by-point suggestions as I believe that the manuscript has potential to reach its final, publishable form just by comprehensively following the general guidelines given above.

Kind regards,

Reviewer

Author Response

We are appreciative for the Reviewer’s careful review and continued suggestions for improving the manuscript. In particular we are grateful for the suggestion of implementing Table 1 as a summary and focus of the Hulk/Deadpool hypothesis. The Reviewer now suggests we use Table 1 as the central focus and delete any text that appears not to feed into this Table. We have made a few such changes in an effort to further streamline the manuscript (e.g. lines 416-422; 466-467; 485-487; 948-951). However, we did not take it to the point of “deleting lengthy parts of text…” as that would constitute a further “major” revision and, in our rewriting of the current version, we didn’t find any such lengthy passages that we felt did not contribute to an important aspect of our message. Moreover, we feel the level of detail is a benefit as it has gone beyond what other similarly focused reviews about cytokinins have provided.

The Reviewer suggests that we delete passages that refer to non-plant organisms (an example is given in lines 906-914). We have taken this suggestion into consideration and shortened some passages (ex. lines 390-391 about agrobacterial Tmr; lines 475-476 about Archeal LOGs or lines 882-887 about Xanthomonas and Mycobacterium signaling) but we did not delete them completely. While we agree that plant processes are the focus of the manuscript we also feel that some comparisons with other, non-plant systems, are warranted. The study of non-plant cytokinin systems is probably the fastest growing area of cytokinin research currently, and in many cases it is not possible or desirable to separate the CK interactions between plants and non-plant interactions (ex. fungi, bacteria, insects, protists). On top of that the study of those other, non-plant systems reveals insight into the primitive pathways and mechanisms that plants either share or from which they have derived further, nuanced systems. Afterall, many aspects of the Cytokinin Universe were first established with help of bacteria (e.g. IPTs, LOGs). The study of non-plant systems, therefore, provides vital clues for plant researchers to act upon.

The reviewer suggested we take out information about molecules “not derived from plant tissues” and the only example given is Kinetin. We respectfully disagree with taking out the kinetin references. Kinetin is engrained in the CK literature historically, analytically and in a few rare cases it is claimed to exist endogenously to plants. To leave it out would have many readers wondering why the paper ignores it, when other aromatic cytokinins – which also have had their endogenous existence questioned – are included. Kinetin also has widespread use as a supplemental, exogenous cytokinin and its effects and metabolism give many clues as to how other cytokinins are produced, taken up, sensed or metabolized. We feel it would make an unnecessary gap in the review to take it out based on some researchers’ opinions that it is not relevant to the Cytokinin Universe.

The Reviewer suggested that we thoroughly revise passages that “sound like criticism of other authors work and/or detailed consideration of scenarios that [were not] taken into account in particular past publications.” One passage in particular was pointed out (lines 1013-28) so we re-edited and significantly shortened that part of the manuscript (lines 988-1007). In hindsight, that area may have been presented as overly-critical; but that certainly was not our intent. However we did not delete that passage entirely as we believe readers should be aware of improvements that should be implemented in future experiments. It is the job of a Review paper to critically appraise past work, but not get overbearing in the process. We believe the new edits find that balance so that future readers will not perceive any misplaced ill-will. We have re-read the manuscript to find other places where the tone could be softened including at the end of Section 6.1 and a passage about glucosyltransferases.

Reviewer 2 Report

I have no further comments on this manuscript.

Author Response

The authors wish to thank the Reviewer for their time invested in our manuscript throughout all round of review.